# YOLOv5s-DSD: An Improved Aerial Image Detection Algorithm Based on YOLOv5s

**DOI:** 10.3390/s23156905

**Published:** 2023-08-03

**Authors:** Chaoyue Sun, Yajun Chen, Ci Xiao, Longxiang You, Rongzhen Li

**Affiliations:** School of Electronic Information Engineering, China West Normal University, Nanchong 637001, China; sunchaoyuepaper@163.com (C.S.); xiaoci035560@163.com (C.X.);

**Keywords:** aerial imagery, object detection, YOLOv5, dense small objects

## Abstract

Due to the challenges of small detection targets, dense target distribution, and complex backgrounds in aerial images, existing object detection algorithms perform poorly in aerial image detection tasks. To address these issues, this paper proposes an improved algorithm called YOLOv5s-DSD based on YOLOv5s. Specifically, the SPDA-C3 structure is proposed and used to reduce information loss while focusing on useful features, effectively tackling the challenges of small detection targets and complex backgrounds. The novel decoupled head structure, Res-DHead, is introduced, along with an additional small object detection head, further improving the network’s performance in detecting small objects. The original NMS is replaced by Soft-NMS-CIOU to address the issue of neighboring box suppression caused by dense object distribution. Finally, extensive ablation experiments and comparative tests are conducted on the VisDrone2019 dataset, and the results demonstrate that YOLOv5s-DSD outperforms current state-of-the-art object detection models in aerial image detection tasks. The proposed improved algorithm achieves a significant improvement compared with the original algorithm, with an increase of 17.4% in mAP@0.5 and 16.4% in mAP@0.5:0.95, validating the superiority of the proposed improvements.

## 1. Introduction

The high-definition, large-scale, and up-to-date characteristics of aerial images captured by unmanned aerial vehicles (UAVs) have made them widely used in various fields such as urban planning [1], security surveillance [2,3], and power line inspection [4,5], demonstrating significant practical value. Traditional methods rely on manual processing to extract information from aerial images, which not only wastes human resources but also introduces human errors in the processed images. With the advancement of computer vision technology, the integration of computer vision with UAVs has greatly improved the utilization efficiency of aerial images. One common approach is to apply object detection techniques to process aerial images. Object detection technology can efficiently and accurately extract key information from images. By incorporating detection techniques into UAVs, tasks such as traffic monitoring, urban planning [6], and resource management [7] can be effectively accomplished, offering immense practical value. The integration of object detection and UAV technology has become a hot research topic among researchers [8,9,10].

However, during the process of applying object detection techniques to aerial image processing, researchers have found that UAV images present challenges such as dense objects, small-sized objects, and complex backgrounds [11]. These issues have hindered the performance of current detection algorithms in aerial image detection tasks. To enhance the detection performance of UAV aerial images and improve the efficiency of their utilization, there is a need to design superior detection algorithms that can address these challenges.

Currently, there are two main categories of object detection techniques for aerial images. The first category is region-based two-stage detection algorithms, represented by R-CNN [12], Fast R-CNN [13], Faster R-CNN [14], and others. The second category is regression-based one-stage detection algorithms, represented by YOLO [15,16,17,18,19,20,21], SSD [22], EfficientDet [23], and others. The former category is characterized by high detection accuracy but slower speed, while the latter category has lower accuracy but faster detection speed. YOLO is one of the representative algorithms for one-stage detection. Through several generations of development, the current YOLO not only maintains its fast detection speed but also achieves comparable accuracy to two-stage detection algorithms. This has attracted the attention of many researchers.

Based on the outstanding performance of YOLO in the current detection algorithms, this paper also chooses YOLO as the base algorithm to propose an improved algorithm called YOLOv5s-DSD to address the aforementioned challenges in aerial image detection. Specifically, to tackle the issues of small objects and complex backgrounds in aerial images, this paper proposes the SPDA-C3 structure, which retains detailed information while enhancing the network’s focus on meaningful features, thereby improving the detection of small objects in complex scenes. The paper designs the Res-DHead detection head structure and adds an additional detection head for detecting tiny objects. By decoupling the classification and localization tasks and fusing multi-scale information, the algorithm successfully improves the detection of small objects. Furthermore, the paper introduces Soft-NMS-CIoU to replace the traditional NMS (non-maximum suppression) method, effectively solving the issue of missing bounding boxes in dense object scenarios. To validate the effectiveness of the proposed algorithm, experiments and comparative tests are conducted on the VisDrone2019 [24] dataset. The results show that the proposed YOLOv5s-DSD model outperforms the original model, achieving a 17.4% improvement in mAP@0.5, and demonstrating superior detection performance compared with current mainstream detection models. The contributions of this paper are as follows:The SPDA-C3 structure is proposed to address the challenges of small objects and complex backgrounds in aerial images in complex scenes.The novel decoupled head, Res-DHead, is introduced and integrated with the detection network, significantly improving the detection performance of the algorithm.The YOLOv5s-DSD model is proposed, which outperforms other current mainstream detection models in aerial image detection tasks.

## 2. Related Works

### 2.1. Feature Extraction

The original image contains a lot of redundant information and noise. Feature extraction can extract meaningful and representative features from the input image, enabling the network to accurately recognize and locate the target [16,17]. Improving the ability of feature extraction allows the network to better capture detailed features of small targets. By learning richer feature representations, the network can better localize and detect various types of targets, enhancing its detection capabilities.

The backbone is the part responsible for feature extraction in the YOLOv5 network, consisting of CBL, CSP, and SPPF layers [18]. The quality of the feature extraction structure has a significant impact on the detection results, and improving the backbone network has become a research focus for many researchers. Many researchers have focused on the attention mechanism, aiming to improve the backbone network by adding additional attention mechanisms to focus on relevant features and enhance the detection performance. For example, Luo X et al. [25] added an IECA attention module after the Focus module, Yao J et al. [26] introduced spatial channel mixed attention mechanism to the backbone network, and Qiu S et al. [27] incorporated coordinate attention mechanism during feature extraction. These researchers have demonstrated through numerous experiments that attention mechanisms can help the network focus on relevant features and enhance feature extraction capabilities.

In addition, improving the residual structure has also been a major research focus for many researchers. Gao S H et al. [28] proposed Res2Net, which introduced a novel Res2Net module that replaces the original residual structure with the goal of multi-scale feature fusion. This module significantly enhances the feature extraction capabilities of the network. In recent years, some researchers have also paid attention to the downsampling operation in the backbone network and the max-pooling operation performed in the SPP layer, which may lead to the loss of detailed information during the process. Detailed information is particularly important for small object detection. Some researchers have proposed new structures to enhance the feature extraction capabilities of the network by reducing the loss of detailed information. For example, He G et al. [29] introduced the SPP-D module, which reduces the feature information loss caused by max-pooling in the SPP module by enhancing the reuse of feature information, thereby improving small object detection capabilities. Sunkara R et al. [30] utilized the SPD-Conv structure to replace the original downsampling structure, successfully reducing information loss and improving the algorithm’s capability to detect small objects.

### 2.2. Detection Head

The detection head is responsible for extracting the position and class information of objects from the processed feature maps and generating the final detection results [16,17]. The performance of the detection head directly affects the detection results. Some researchers have proposed their own improvement methods for the detection head and achieved certain success, among which the decoupled head has performed exceptionally well.

The detection head used in the YOLOv5 algorithm is a coupled head, which combines the classification and localization tasks together for processing. However, in reality, these two tasks have different focuses, and coupling them together inevitably leads to performance loss. To address this issue, YOLO X [31] proposed the decoupled head. The decoupled head decouples the classification and regression tasks, allowing each task to focus on the necessary information, thus improving the detection performance of the algorithm. Additionally, researchers such as Lu S [32] and Li C [19] proposed variants of decoupled heads, such as asymmetric decoupled head and efficient decoupled head, further demonstrating the effectiveness of the decoupling approach.

Another common approach for improving the detection head is to add an additional head specifically designed for detecting small objects to the existing three detection heads in YOLOv5. Researchers such as Baidya R [33] and Zhao Q [34] have adopted this approach and have extensively demonstrated its effectiveness in improving the network’s ability to detect small objects through numerous experiments.

### 2.3. NMS

After obtaining the feature maps processed by the algorithm, post-processing is required to obtain the final detection results. During the inference process of object detection, the algorithm generates multiple bounding boxes, each with a confidence score. To complete the detection task, it is necessary to eliminate useless boxes and select the optimal ones [17,18]. The filtering process consists of two steps: firstly, a large number of invalid boxes are removed based on a set confidence threshold, and then NMS is used to eliminate redundant boxes that detect the same object. Finally, the optimal boxes are obtained.

The NMS used in YOLOv5 performs the filtering of unique boxes by calculating the IoU (Intersection over Union) [35] between the box with the highest current score and other boxes to determine if surrounding boxes are redundant detections of the same object. However, this method tends to incorrectly classify useful boxes detecting neighboring objects as redundant, leading to missed detections when dealing with dense objects. Many researchers have proposed their own improvement methods for this issue. For example, He G et al. [29] used DIoU-NMS to replace the original NMS, successfully improving the detection accuracy of small objects in complex backgrounds. Zhang H et al. [36] combined Cluster-NMS [37] with DIoU-NMS to propose the Cluster-DIoU-NMS method, which solves both over-suppression and overlapping issues.

## 3. Methods

This article proposes an improved algorithm, YOLOv5s-DSD, based on YOLOv5s, addressing the challenges faced in aerial image detection from three aspects: feature extraction, detection head improvement, and post-processing. Firstly, the SPDA-C3 structure is designed to help the algorithm retain crucial detail information that is easily lost during downsampling. By replacing the original C3 structure in the backbone with SPDA-C3, the feature extraction capability of the algorithm is enhanced, leading to improved detection performance for small objects. Secondly, for the detection head, the Res-DHead structure is designed, and an additional specialized detection head for detecting small objects is added. The proposed improvements not only enhance the network’s ability to detect small objects but also decouple the tasks of localization and classification in the detection head. This allows the detection head to fully utilize the obtained information, improving overall detection performance. Finally, the Soft-NMS-CIoU structure is used to replace the original NMS structure, addressing the issue of missed detections caused by densely packed objects. The overall network structure of the improved YOLOv5s-DSD algorithm is shown in Figure 1.

### 3.1. Replace Backbone Network C3 with SPDA-C3

The backbone network of YOLOv5 primarily consists of alternating CBL and C3 structures. The CBL layer, with a stride of two in the backbone network, is responsible for feature extraction and downsampling operations. However, using CBL for downsampling inevitably leads to information loss. The C3 structure, on the other hand, primarily focuses on increasing the depth of the network and enhancing feature extraction capabilities. This paper proposes a novel structure called SPDA-C3 based on the C3 structure. Figure 2 illustrates the overall structure of the C3 in the original model’s backbone network.

The SPDA-C3 structure is composed of space to depth, CA, Conv, and other components. SPDA-C3 not only possesses the ability of the original C3 structure to increase network depth and improve feature extraction, but it also performs downsampling operations without information loss. Among them, space to depth is a spatial deepening layer, and its working principle is illustrated in Figure 3. It extracts features by skipping one pixel in both the rows and columns of the original feature map. This process generates four new feature maps, which are then concatenated along the channel dimension. As a result, a new feature map is obtained with four times the number of channels as the original feature map, but with half the resolution.

By incorporating the coordinate attention mechanism after the space-to-depth layer, the network’s focus on relevant information in the extracted feature maps can be further enhanced. Following that, a convolution layer with a kernel size of 3 × 3 and a stride of one is used for final feature extraction, allowing for downsampling operations without information loss. The subsequent structure of SPDA-C3 retains the original structure of C3, enabling the module to balance the tasks of increasing network depth and feature extraction. The overall structure of SPDA-C3 is illustrated in Figure 4.

It should be noted that the SPDA-C3 structure replaces the downsampling function of CBL. Therefore, when replacing the original C3 structure in the backbone network with the SPDA-C3 structure, the stride of the CBL layer with a stride of two in the backbone network needs to be changed to one to avoid redundant downsampling operations. Additionally, in order to further minimize information loss, this paper chooses to replace the first CBL structure of the original backbone network with a focus module.

### 3.2. RES-DHead Structure

Inspired by the Res2Net network and the decoupled head idea, this paper proposes the Res-DHead structure and adds an additional detection head for detecting small objects on top of the original detection head to improve the algorithm’s performance in detecting small objects. The Res-DHead structure consists of C3-Res2 and the decoupled head, as shown in Figure 5.

The C3-Res2 structure combines the Res2Net module with the C3 structure by replacing the original residual structure with the Res2Net structure. The Res2Net module performs multi-scale fusion on the input feature maps, allowing the network to utilize richer feature information. The specific structure is shown in Figure 6. The improved C3-Res2 structure performs additional multi-scale fusion on the feature maps before they are passed to the detection head, enabling the detection head to utilize more abundant feature information. The C3-Res2 structure is illustrated in Figure 7.

The decoupled head extracts localization and class information from the obtained feature map in the C3-Res2 structure. The decoupled head uses two prediction branches to separately predict the classification and localization tasks. The overall processing flow of the decoupled head is shown in Figure 8. The classification branch utilizes two 3 × 3 convolutional layers for processing and outputs the category information. The localization branch generates two branches after being processed by two convolutional layers, obtaining confidence score predictions and regression box predictions, respectively. Compared with traditional detection heads, the decoupled head decouples classification and regression, allowing each task to focus on its required information, effectively improving the detection performance of the algorithm.

The Res-DHead structure is designed by incorporating the Res2Net concept and decoupling idea. By integrating multi-scale information, the receptive field is enhanced, allowing the algorithm to better utilize both detailed and semantic information. Considering the different focuses of the classification and localization tasks, the decoupling of these tasks enables them to better attend to the information specific to each task. Subsequent experiments have successfully validated that the addition of the Res-DHead detection head improves the algorithm’s detection performance.

### 3.3. Soft-NMS-CIoU Replaces NMS

Aerial images captured by unmanned aerial vehicles (UAVs) are often taken from high altitudes, resulting in the issue of dense object detection. In such cases, the predicted bounding boxes for different objects tend to overlap. Traditional non-maximum suppression (NMS) can mistakenly eliminate bounding boxes that detect different objects but are close in proximity, leading to poor detection performance for dense targets. The working principle of Soft-NMS is different from NMS. Soft-NMS [38] calculates the IoU of the highest scoring box and the bounding box and suppresses the bounding box to varying degrees based on the level of IoU, rather than directly setting the scores of other boxes to 0 for deletion. This approach effectively reduces the problem of losing neighboring object prediction boxes due to dense target scenarios. The overall formula for Soft-NMS is shown in Equation (1):(1)$$s_{i}=\left\{\begin{array}{ll} s_{i}, & \mathrm{iou}\;\left(M,b_{i}\right)<N_{t}\\ s_{i}\left(1-\mathrm{iou}\;\left(M,b_{i}\right)\right), & \mathrm{iou}\;\left(M,b_{i}\right)\geq N_{t} \end{array}\right.$$

It should be noted that Soft NMS needs to be calculated through IoU, so the selection of loss function will have a great impact on the suppression effect of Soft NMS. The original Soft NMS uses the IoU function based on the original Soft NMS. This loss function has problems such as a slow regression speed when boxes contain each other and an inability to reflect the distance between boxes when two boxes do not intersect. In comparison with other IoU loss functions, CIoU [39] overcomes the issue of gradient vanishing when two boxes do not intersect and resolves the problem of GIoU [40] degenerating into IoU when two boxes are contained within each other. CIoU inherits the advantages of DIoU while also considering the aspect ratio of the bounding boxes. These advantages enable CIoU to better address the issue of overlapping bounding boxes. Based on this, this paper proposes an improved NMS method by replacing IoU with CIoU, resulting in the enhanced NMS method called Soft-NMS-CIoU. The formula for calculating CIoU is as follows:(2)$$L_{CIoU}=1-IoU+\frac{\rho^{2}\left(b,b^{gt}\right)}{c^{2}}+\alpha v$$
(3)$$R_{CIoU}=\frac{\rho^{2}\left(b,b^{gt}\right)}{c^{2}}+\alpha v$$
(4)$$\alpha =\frac{v}{1-IoU+v}$$
(5)$$v=\frac{4}{\pi^{2}}{\left(\arctan\frac{w^{gt}}{h^{gt}}-\arctan\frac{w}{h}\right)}^{2}$$

In the formula, c is the diagonal length of the minimum bounding box covering two boxes, w and h represent the width and height of the predicted box, $w^{gt}$ and $h^{gt}$, the width and height of the real box, and b and $b^{gt}$ represent the center of the predicted bounding box and the real bounding box point. ρ represents the Euclidean distance between b and $b^{gt}$, and IoU represents the intersection-over-union ratio. α is a positive trade-off parameter, and v is an aspect ratio consistency parameter.

The proposed Soft-NMS-CIoU combines the ideas of Soft-NMS and CIoU, addressing both the issue of IoU calculation for overlapping boxes and the problem of neighboring box loss. To verify the effectiveness of the proposed method, this paper conducts experimental comparisons by combining Soft-NMS with multiple loss functions such as GIoU, EIoU [41], SIoU [42], etc. The experimental results demonstrate that Soft-NMS-CIoU outperforms the other methods, confirming the effectiveness of the proposed approach.

## 4. Experiments

### 4.1. Dataset Description

The VisDrone2019 dataset was created by Tianjin University and consists of aerial images captured by unmanned aerial vehicles (UAVs). The dataset contains a total of 10,209 images, including 6471 training images, 548 validation images, and 3190 test images. It encompasses ten annotated object categories: pedestrians, people, buses, cars, vans, trucks, bicycles, tricycles with sunshades, motorcycles, and tricycles. Figure 9 presents the distribution of category counts, spatial locations, and sizes within the dataset, indicating the presence of numerous small objects and dense object arrangements. Figure 10 showcases a selection of images from the dataset.

### 4.2. Implementation Details

The VisDrone 2019-DET-train dataset was chosen as the training set, while the VisDrone 2019-DET-val dataset was used as the validation set. During training, the initial learning rate was set to 0.01 and the final learning rate was set to 0.1. The classification categories were set to 10, the batch size was set to 10, and the number of epochs was set to 300. The image size was set to 640.

All experiments were conducted on NVIDIA RTX 4090 GPU devices, and the deep learning framework used was PyTorch 2.0.1.

### 4.3. Evaluation Metrics

The evaluation metrics for the experimental results include precision, recall, mAP@0.5, and mAP@0.5:0.95. The focus is particularly on the comprehensive metrics of mAP@0.5 and mAP@0.5:0.95. The formulas for all evaluation metrics are as follows:(6)$$P=\frac{TP}{TP+FP}$$
(7)$$R=\frac{TP}{TP+FN}$$
(8)$$mAP=\int_{0}^{1}P(R)dR$$

In the formulas, *TP* represents the number of true positive samples, *FP* represents the number of false positive samples predicted as true by the algorithm, and *FN* represents the number of missed detections.

### 4.4. Comparison Experiments of Soft-NMS

To further test the performance improvement of Soft-NMS and the impact of different loss functions on Soft-NMS, this study conducted comparative experiments using different methods including traditional NMS, Soft-NMS-CIoU, Soft-NMS-GIOU within the YOLOv5s algorithm framework. The experimental results are shown in Table 1.

Based on the analysis of the data, traditional NMS performs poorly on dense datasets because it mistakenly considers neighboring boxes detecting other objects as redundant detections of the same object. This leads to the erroneous removal of neighboring boxes, resulting in a decrease in model recall and a subsequent drop in overall performance. Soft-NMS, through smoothing neighboring boxes, avoids the incorrect removal of neighboring boxes, and replacing traditional NMS with Soft-NMS significantly improves detection performance. Moreover, different loss functions can affect the performance of Soft-NMS. 

CIoU addresses the issue of gradient disappearance in IoU when two boxes do not intersect and also resolves the problem of GIoU degrading to IoU when two boxes are contained within each other. It inherits the advantages of DIOU and takes into account the aspect ratio of bounding boxes, making CIoU more effective in assisting Soft-NMS in dealing with overlapping bounding boxes. Experimental results once again demonstrate that CIoU, as a loss function for Soft-NMS, exhibits superior performance, further validating the proposed theory in this paper.

### 4.5. Ablation Study

This paper conducts ablation experiments using the YOLOv5s algorithm as the base framework and the VisDrone2019 dataset. The main evaluation metrics are precision, recall, mAP@0.5, and mAP@0.5:0.95.The results of ablation experiments are shown in Table 2.

SPDA-C3: After replacing the backbone network C3 with the SPDA-C3 structure, all four metrics of the algorithm improved. The original YOLOv5s algorithm uses a convolution with a stride of two for downsampling, which leads to a significant loss of fine-grained information. The feature information of small objects is often contained within these lost fine-grained details. Therefore, the detection task for small objects requires the algorithm to fully utilize the detailed information. The SPDA-C3 structure decomposes the original feature map into four smaller feature maps and concatenates them along the channel dimension, allowing the model to perform downsampling while fully utilizing all feature information. This effectively reduces the loss of detailed information and utilizes the CA module to focus on essential features. In general, compared with traditional downsampling methods, using SPDA-C3 to perform downsampling allows the network to fully utilize fine-grained information, thereby enhancing the detection performance of small objects. The results of the ablation experiments above successfully verify that the SPDA-C3 structure brings significant improvements to small object datasets.

RES-DHead: After replacing the original detection head structure with the RES-DHead and adding an additional layer for detecting tiny objects, all four metrics of the algorithm show improvements. The original YOLOv5s algorithm uses a coupled head that combines classification and regression tasks. However, classification and regression have different focus areas in the feature space, and forcefully coupling these tasks together can lead to a certain degree of performance degradation for both classification and regression, resulting in a compromised overall network performance. With RES-DHead, the classification and regression tasks are separated, allowing each task to fully utilize the feature information and focus on their respective areas of interest, which enhances the detection performance of the network. Additionally, compared with the original algorithm’s detection head structure, RES-DHead further conducts multi-scale fusion on the feature maps, enabling the network to leverage more comprehensive feature information.

Furthermore, the additional RES-DHead specifically designed for detecting tiny objects directs the network’s attention to the feature maps with rich shallow-level information, effectively improving the model’s performance in detecting small objects. Ultimately, the results of the ablation experiments demonstrate that after using RES-DHead to replace the original detection head structure and adding an extra RES-DHead dedicated to detecting tiny objects, the overall performance of the network significantly improved. These results strongly validate the effectiveness of the RES-DHead structure.

Soft-NMS-CIoU: After replacing the original NMS with Soft-NMS-CIoU, all four metrics show significant improvements. In the presence of a large number of dense objects, the prediction boxes for different objects may be too close, making NMS prone to mistakenly consider these prediction boxes as duplicates of the same object and delete them, leading to a significant number of missed detections. Soft-NMS-CIoU addresses this issue by applying a smoothing process to neighboring boxes. Specifically, Soft-NMS-CIoU suppresses the neighboring boxes to varying degrees based on the computed CIoU values instead of directly deleting them. This allows the correctly predicted boxes that were mistakenly deleted to be preserved, significantly reducing the number of missed detections caused by erroneous deletion of neighboring boxes.

After replacing the original NMS with Soft-NMS-CIoU, the mAP@0.5 increased by 4.4%. This experimental result validates our approach: compared with traditional NMS, Soft-NMS-CIoU performs better on datasets with dense objects.

To visually demonstrate the significant improvement of YOLOv5s-DSD compared with the original algorithm, we use Figure 11 to showcase its performance. The four images at the top represent the detection results of YOLOv5s-DSD, while the images below display the results of YOLOv5s original algorithm. It is evident that YOLOv5s-DSD accurately detects small objects that were missed by YOLOv5s.

Moreover, YOLOv5s-DSD achieves an FPS of 40FPS during detection, fully meeting the real-time detection requirements. Additionally, compared with the original YOLOv5s model, YOLOv5s-DSD exhibits a significant leap in detection performance.

### 4.6. Comparison of Different Detectors

To validate the effectiveness of the proposed improved algorithm YOLOv5-DSD, this paper compares it with several popular single-stage and two-stage detection algorithms. All algorithms were trained for 300 epochs, and the VisDrone2019 dataset was chosen for testing and comparison using mAP@0.5 and mAP@0.5:0.95 as the evaluation metrics. All the results are shown in Table 3. The results indicate that the proposed YOLOv5s-DSD algorithm achieves the best performance in aerial image detection tasks. Even compared with the latest YOLOv8 detection model, YOLOv5s-DSD outperforms it by 9.5% in mAP@0.5. Additionally, compared with the original model, the improved model shows a 17.4% increase in mAP@0.5.

## 5. Conclusions

To address the challenges in aerial image detection, improve the detection performance of aerial images, and better utilize aerial images, this paper proposes an improved algorithm called YOLOv5s-DSD based on YOLOv5s. The improved algorithm utilizes the SPDA-C3 structure to make the network focus on important information, solving the problem of complex backgrounds in aerial images. Additionally, the SPDA-C3 structure reduces information loss during the downsampling process, preserving detailed information and significantly improving the algorithm’s ability to detect small objects. The proposed Res-DHead structure combines multi-scale information fusion and task decoupling, enabling the algorithm to fully utilize feature information and greatly enhance overall detection performance. To address the issue of misclassification caused by densely packed objects, the original NMS method is replaced with Soft-NMS-CIoU, reducing the occurrence of false negatives due to dense objects.

Testing on the VisDrone2019 aerial dataset confirms the effectiveness of the proposed improved algorithm design. The proposed YOLOv5s-DSD achieves a 17.4% improvement in mAP@0.5 compared with the original model. Even when compared with the state-of-the-art YOLOv8s, YOLOv5-DSD still outperforms it by 9.5% in mAP@0.5. Experimental results demonstrate that YOLOv5s-DSD exhibits superior detection performance for aerial images compared with the current mainstream detection models.

In future work, efforts will be made to address the limitations of the proposed method in the paper. Considering the importance of convenience in unmanned aerial vehicle (UAV) devices, algorithm models with lower computational requirements are more suitable for UAV deployment. Therefore, future research will focus on the issue of lightweighting the algorithm, aiming to further reduce the number of parameters and enable the algorithm to be deployed on UAV devices with lower computational capabilities.

## Figures and Tables

**Figure 1 sensors-23-06905-f001:**
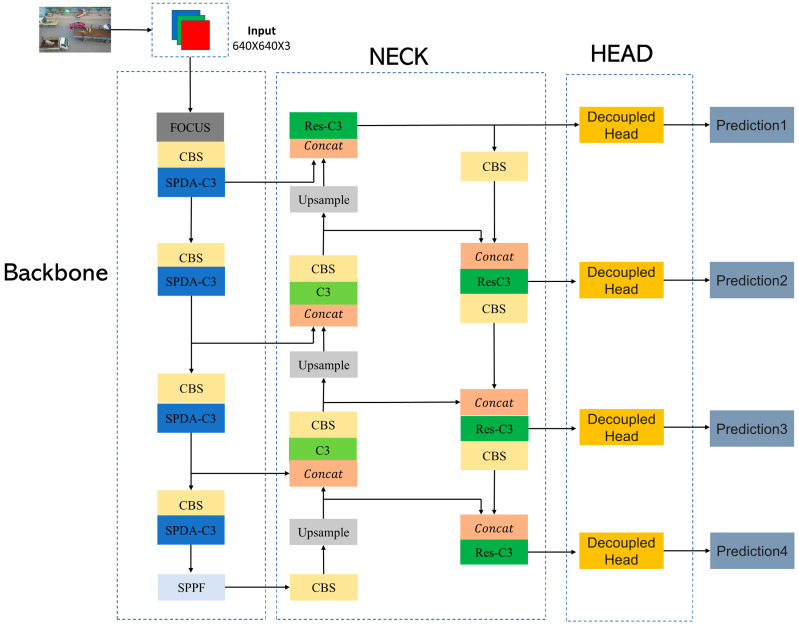
YOLOv5s-DSD network configuration.

**Figure 2 sensors-23-06905-f002:**
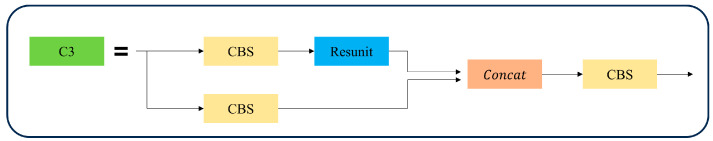
C3 structure in the original model’s backbone network.

**Figure 3 sensors-23-06905-f003:**
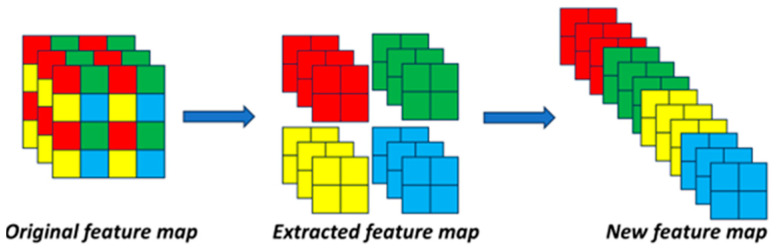
Diagram of the space-to-depth principle.

**Figure 4 sensors-23-06905-f004:**
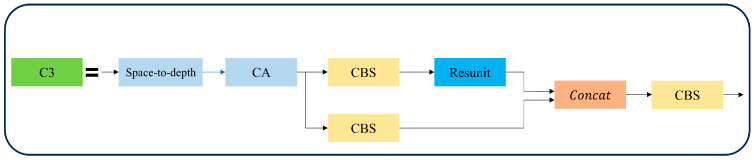
SPDA-C3 structure.

**Figure 5 sensors-23-06905-f005:**
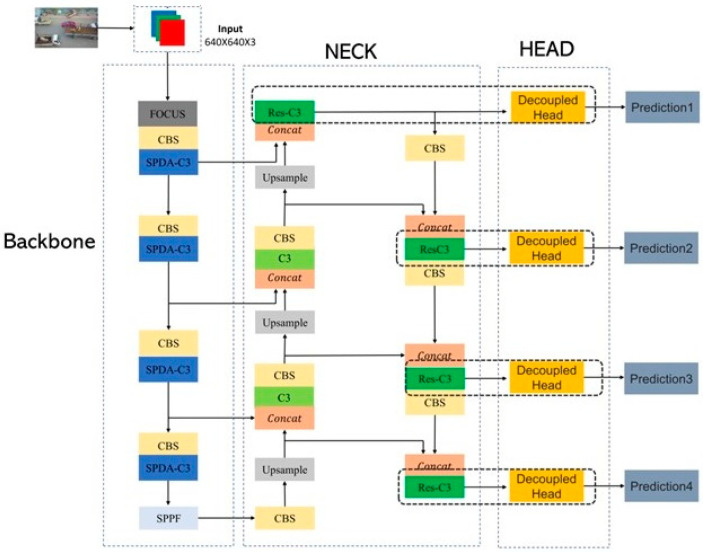
Res-DHead structure.

**Figure 6 sensors-23-06905-f006:**
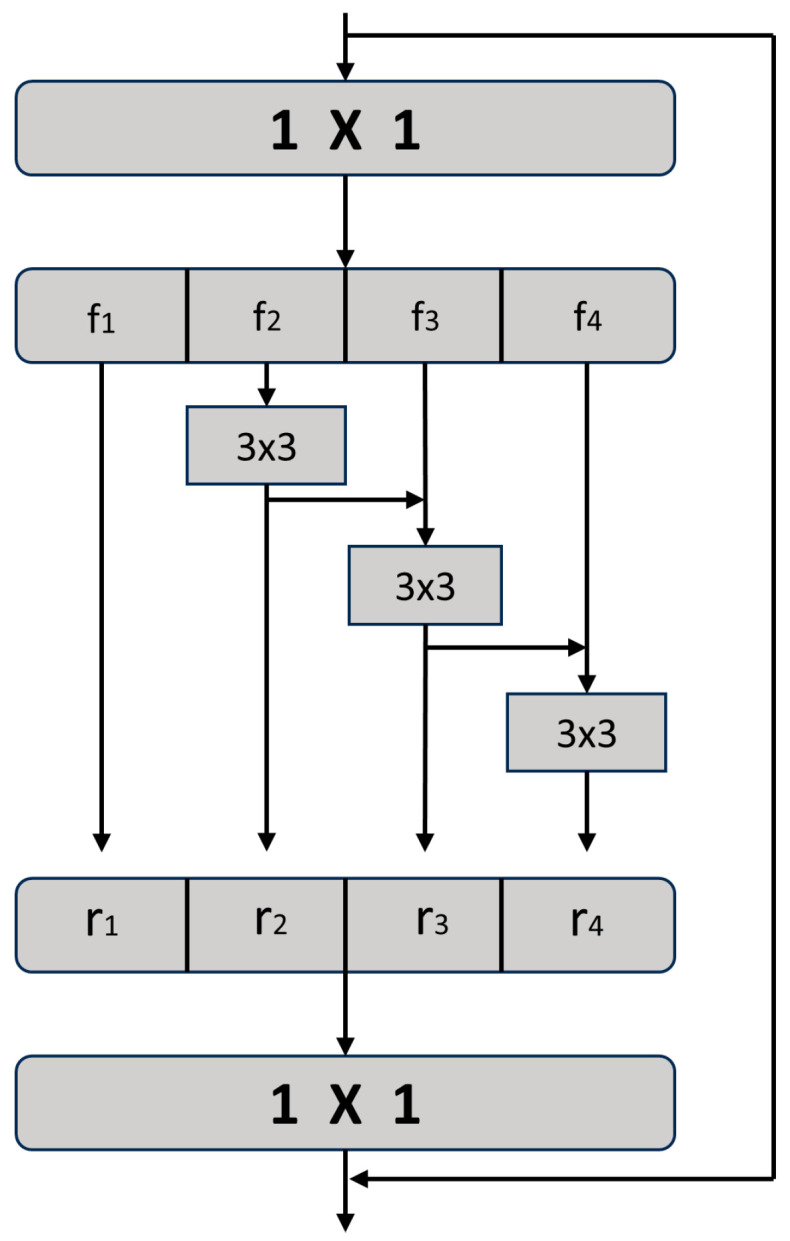
Res2Net module structure.

**Figure 7 sensors-23-06905-f007:**
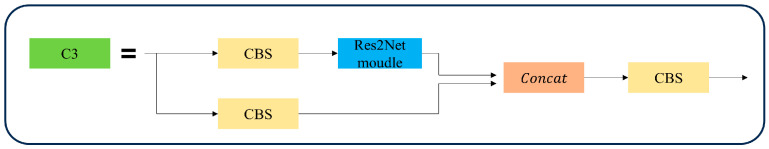
Overall structure of C3-Res2.

**Figure 8 sensors-23-06905-f008:**
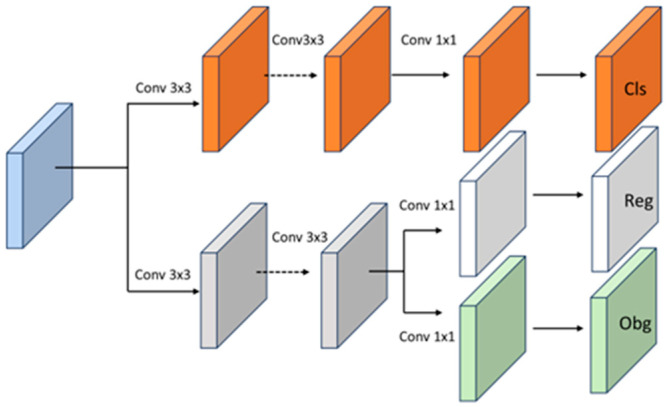
Decoupled Head Flowchart.

**Figure 9 sensors-23-06905-f009:**
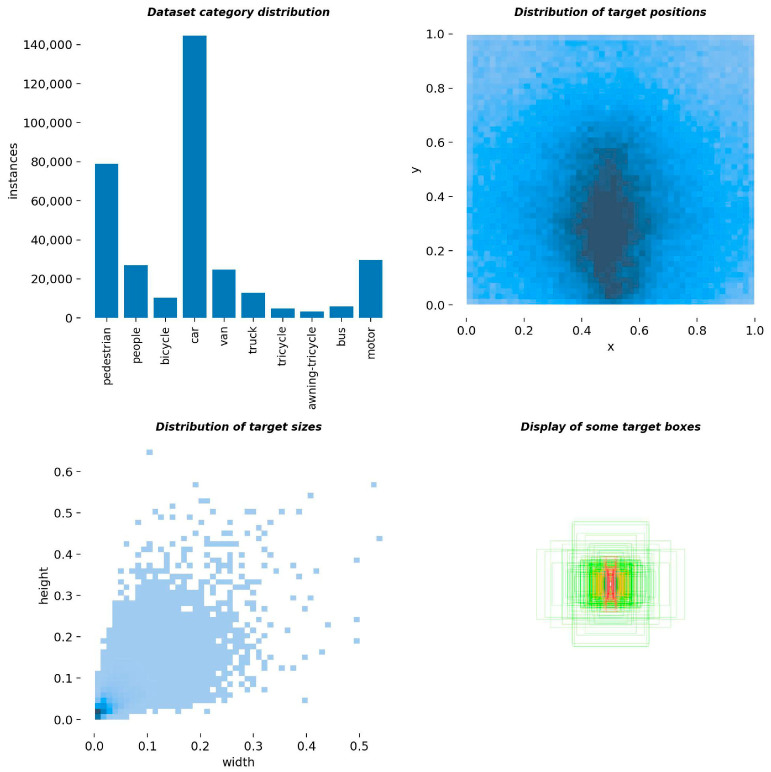
VisDrone dataset label information.

**Figure 10 sensors-23-06905-f010:**
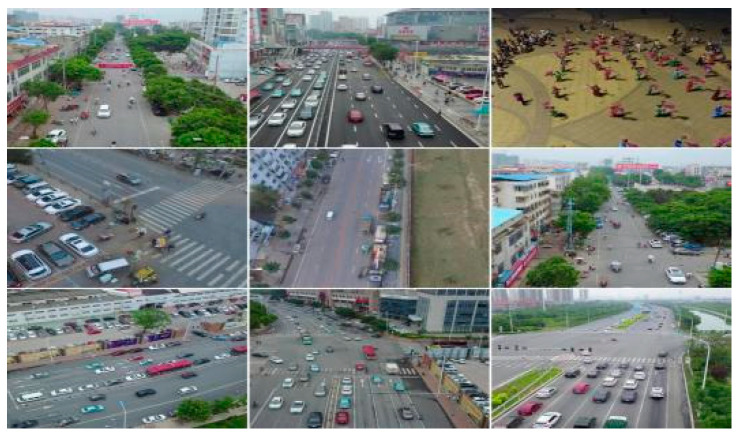
Partial dataset showcase.

**Figure 11 sensors-23-06905-f011:**
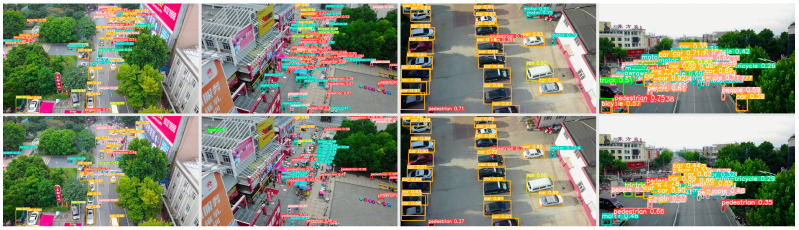
Detection result showcase.

**Table 1 sensors-23-06905-t001:** Comparative experiments of traditional NMS and Soft-NMS with different loss functions. The best-performing methods are highlighted in bold.

Method	P	R	mAP@0.5	mAP@0.5:0.95
NMS	46.3	33.5	33.5	18.0
Soft-NMS-IoU	52.4	28.2	40.4	24.4
Soft-NMS-GIoU	53.0	29.4	41.2	24.9
Soft-NMS-DIoU	53.6	29.3	41.6	25.3
Soft-NMS-SIoU	52.5	28.9	41.3	25.3
Soft-NMS-EIoU	**58.4**	20.3	39.7	24.6
**Soft-NMS-CIoU (ours)**	53.7	**29.3**	**41.7**	**25.5**

**Table 2 sensors-23-06905-t002:** Ablation study results.

Method	P	R	mAP@0.5	mAP@0.5:0.95
Baseline	46.3	33.5	33.5	18.0
YOLOv5s+SPDA-C3	44.6	37.0	35.2	19.1
YOLOv5s+SPDA-C3+RES-DHead	55.1	44.3	46.5	27.2
YOLOv5s+SPDA-C3+RES-DHead+Soft-NMS-CIoU	57.4	44.8	50.9	32.4

**Table 3 sensors-23-06905-t003:** Comparison of results on the VisDrone2019 dataset with different algorithms. The best result is indicated in bold.

Method	mAP@0.5	mAP@0.5:0.95
SSD [22]	10.6	5.0
EfficientDet [23]	21.1	12.8
RetinaNet [43]	25.6	15.1
CenterNet [44]	29.1	14.0
Faster R-CNN [14]	35.6	19.6
YOLOv3-SPP [16]	18.8	10.6
YOLOv5s [18]	33.5	18.0
YOLO-UAVlite [45]	36.6	20.6
KPE-YOLOv5s [46]	39.2	22.4
YOLOv7 [20]	37.4	23.8
YOLOv8s [21]	41.3	24.9
**YOLOv5s-DSD (ours)**	**50.9**	**32.4**

## Data Availability

Not applicable.

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
