# Peer review of "YOLOv5s-DSD: An Improved Aerial Image Detection Algorithm Based on YOLOv5s"

_sensors, 2023, doi:10.3390/s23156905_

Round 1

Reviewer 1 Report

1.        The idea of the work is interesting. Improving YOLOv5 for aerial image detection to tackle the challenges of small detection targets, dense target distribution, and complex backgrounds. It can also significantly improve compared to the original algorithm, with an increase of 17.4% in [email protected] and 16.4% in [email protected]:0.95 in the VisDrone2019 dataset. Why you don’t utilize more other relevent datasets? You need to compare your approach in other aerial imagery datasets to prove your approach. If it can’t be established, you need to explain the reason.

2.        The VisDrone2019 dataset seems so challanging and suitable for your work for aerial imagery object detection. In your work, you utilized a total of 10,209 images, including 6,471 training images (63.39%), 548 validation images (5.37%), and 3,190 test images (31.25%). Why did you use that composition (training: validation: test = 63.39: 5.37: 31.25)? is it fundamentally approved to the work? You need to provide references to approve it.

3.        What happened in Table 3 (“Error! Reference source not found.”)?. Anyway it is good point you compare also with YOLO-UAV and also the latest versions of YOLO: YOLO 6, 7, and 8.

4.        “With the advancement of computer vision technology, the integration of computer vision with UAVs has greatly improved the utilization efficiency of aerial images. One common approach is to apply object detection techniques to process aerial images. Object detection technology can efficiently and accurately extract key information from images. By incorporating detection techniques into UAVs, tasks such as traffic monitoring, urban planning, and resource management can be effectively accomplished, offering immense practical value. The integration of object detection and UAV technology has become a hot research topic among researchers.” (first paragraph in Section 1. Introduction) à you need references to support your statements.

5.        “However, during the process of applying object detection techniques to aerial image processing, researchers have found that UAV images present challenges such as dense objects, small-sized objects, and complex backgrounds. These issues have hindered the performance of current detection algorithms in aerial image detection tasks.” (second paragraph in Section 1. Introduction) à you need references to support your statements.

6.        “The original image contains a lot of redundant information and noise. Feature extraction can extract meaningful and representative features from the input image, enabling the network to accurately recognize and locate the target. Improving the ability of feature extraction allows the network to better capture detailed features of small targets. By learning richer feature representations, the network can better localize and detect various types of targets, enhancing its detection capabilities.” (first paragraph in Section 2.1. Feature extraction) à you need references to support your statements.

7.        “The backbone is the part responsible for feature extraction in the YOLOv5 network, consisting of CBL, CSP, and SPPF layers.” (second paragraph in Section 2.1. Feature extraction) à you need references to support your statements. Maybe you can refer to the original paper of YOLOv5.

8.        “The detection head is responsible for extracting the position and class information of objects from the processed feature maps and generating the final detection results. The performance of the detection head directly affects the detection results.” (first paragraph in Section 2.2. Detection head) à you need references to support your statements.

9.        “After obtaining the feature maps processed by the algorithm, post-processing is required to obtain the final detection results. During the inference process of object detection, the algorithm generates multiple bounding boxes, each with a confidence score. To complete the detection task, it is necessary to eliminate useless boxes and select the optimal ones. The filtering process consists of two steps: firstly, a large number of invalid boxes are removed based on a set confidence threshold, and then non-maximum suppression (NMS) is used to eliminate redundant boxes that detect the same object. Finally, the optimal boxes are obtained.” (first paragraph in Section 2.3. NMS) à you need references to support your statements.

10.     Please mention your results in Section 5. Conclusions.

-

Author Response

Dear Reviewer,

Thank you very much for your valuable feedback. We have completed all the necessary revisions as per your suggestions. Additionally, we would like to address your inquiries:

Regarding your first question: Why did we not use other datasets for performance evaluation?

There are two main reasons for this decision. Firstly, our laboratory has limited resources, with only one computer available for conducting public experiments. With limited available devices, using additional datasets would strain our resources and consume a significant amount of time, considering that other members in the lab have their own experiments to conduct as well. Secondly, after thorough investigation, we found VisDrone2019, a high-quality aerial drone dataset, which poses significant challenges. Moreover, this dataset is widely recognized and respected within the research community. As evidenced by the information we presented about the dataset, we believe it is well-suited for evaluating the effectiveness of our proposed experimental methods. Due to these reasons, we did not use additional datasets, and we hope for your understanding.

Regarding your second question: Why is our dataset distribution ratio 63.39:5.37:31.25?

This distribution ratio is provided by the official VisDrone2019 dataset for conducting object detection challenge tasks. The ratio has been acknowledged by the dataset creators, and its usage has been widely adopted by numerous researchers to effectively assess model performance. Hence, we decided to maintain this distribution ratio for consistency and compatibility with existing research.

Regarding your third point, we have rectified the reference links in Table 3.

Regarding points 4 to 9, we have added corresponding references to reinforce the credibility of our statements.

Regarding point 10, we have included additional explanations about the results in Section 5. Conclusions.

Once again, we appreciate your invaluable feedback, and all the revised sections are highlighted in blue for easy identification.

Sincerely
Mr.Sun

Reviewer 2 Report

The paper proposes YOLOv5s-DSD algorithm on the basis of YOLOv5s, which is innovative, but the experimental analysis is not sufficient, the specific recommendations are as follows.

1. increase the visualization of detection results.

2. Increase the analysis of the efficiency of the network.

3. increase the comparison experiment with other literature methods.

4. the analysis of experimental content needs to be more detailed.

There are no obvious grammatical errors in the English language, but some textual expressions need to be touched up to improve fluency.

Author Response

Dear Reviewer,

Thank you very much for your valuable feedback. We have made the necessary revisions as per your suggestions:

1. We have showcased the detection results of our improved algorithm and compared them with the original algorithm to highlight its effectiveness.

2. Additionally, we have added extra efficiency analysis.

3. We have resolved the issue with the comparison section and now Table 3 displays the comparison results between our algorithm and other state-of-the-art algorithms.

4. We have conducted a more in-depth analysis of the experiments.

Once again, we sincerely appreciate your valuable feedback, and all the corresponding revisions have been highlighted using red color markers.

Sincerely
Mr.Sun

Reviewer 3 Report

1. Summary

Presented article introduces extension/modification of YOLOv5s algorithm for small object detection for aerial images. Based on introduced requirements, the new, improved architecture of YOLOv5s model is presented. Modifications of YOLOv5s and contribution of article is in three areas: 1) improved YOLOv5s backbone model architecture for better small objects detection, 2) design of the new Decoupled Head, 3) experimental evaluation on VisDrone2019 dataset. The article provides very good overview of all steps and explains methodology and experimental evaluation.

2. Citations and resources

Cited references are relevant. Article contains some broken links to references (Table 3).

3. Manuscript

The article has good structure, it is very good for reading, and provides all necessary details for problem, methodology, and application understanding. Experimental design follows standards defined for similar types of research and provides results in clear structure. All the necessary evaluation criteria are well described and presented inside the article.

3.1 General comments

Based on review criteria, I have following comments to submitted article:

I recommend using explanation of all the abbreviations in the text when first used. Some of them are generally known (e.g., IoU) but using whole name can improve the readability (explanation) and avoid misunderstanding.

(L: denotes the line number)

1. L:265-270

In the formula for CIOU loss (formula 2 and 3), the capital C is used. Following the common definition, c is used in the formulas which denotes diagonal length of the smallest enclosing box covering the two boxes (C denotes smallest box). Please, correct the formulas. Additionally, there should be explanation of all the variables used in the formulas!

2. L:322 – Table 2

There is no precision column in the table, even it is mentioned in the text. Please, correct the table.

3. L:357 – Table 3

All the references are broken in provided manuscript for the review. Please, solve the problem with references.

3.2 Specific comments

1. L:69

There is redundant reference to VisDrone2019 dataset.

2. L:285 – Figure 9

I recommend to add captions for each subplot for quicker information.

4. Reproducibility

Experimental evaluation is defined for generally available VisDrone2019 dataset, and all the algorithms used in comparison are referenced.

Author Response

Dear Reviewer,

Thank you for your valuable feedback. We have completed all the corresponding revisions as per your suggestions. Below is a brief summary of the modifications:

1. We have added the full names for IOU and NMS when they first appear in the manuscript.
2. Addressing your concern regarding the CIoU formula, we have made the necessary modifications and added explanations for all the symbols in the formula.
3. We have included the accuracy column in Table 2. Additionally, we have added full names for IOU and NMS when they are first mentioned in the manuscript.
4. The issue with broken reference links in Table 3 has been fixed.
5. The redundant reference to VisDrone has been removed.
6. We have provided corresponding subtitles for all the subplots in Figure 9.

Once again, we sincerely appreciate your valuable feedback, and all the improvements have been highlighted using yellow color markers.

Sincerely
Mr.Sun

Round 2

Reviewer 2 Report

This paper proposes YOLOv5s-DSD algorithm on the basis of YOLOv5s, which is innovative. After modification, its experimental analysis is more adequate and agreed to be accepted.

This paper still needs a little editing in English.